# Limitations in Modelling Reinforced Concrete Durability

Chris Atkins [1] and Paul Lambert [2,*]

1 Technical Principal, Materials and Corrosion Technology, Mott MacDonald, Altrincham WA14 1ES, UK; chris.atkins@mottmac.com
2 Centre for Infrastructure Management, Sheffield Hallam University, Sheffield S1 1WB, UK
* Correspondence: p.lambert@shu.ac.uk

**Abstract:** The design processes for reinforced concrete are changing. More often, durability targets are being achieved by using modelling. This paper compares some of the models available and the precision undertaken to obtain the data that underpins the calculations, and it reflects on the change in the environment that is known to be occurring. In addition, a review of the sustainability implications of durability is considered. It is concluded that there may be more sustainable methods to achieve a long life than simply increasing cement contents and covers.

**Keywords:** modelling; durability; concrete; sustainability





## 1. Introduction

For reinforced concrete, the design process is changing. Longer design lives are being sought on major projects. In addition, sustainability requirements are being introduced. This is typically measured by calculating the carbon dioxide released to the environment as a result of the project during its whole life cycle, referred to as the embodied carbon. The aim is to drive down the embodied carbon for each project. The main contributor in reinforced concrete is often considered to be cement. Increased durability often demands an increased cement or cementitious material content, either directly or indirectly, as a lower water/cement material can result in a need to increase the cement content to maintain the fluidity of a mix. Reducing embodied carbon often demands a decrease in cement content. This can lead to a conflict between a durability modelling and the sustainability requirement. While it is common to think the main durability concern is the corrosion of reinforcement, which is initiated by chloride ingress or carbonation, aggressive ground can also contribute.

Modelling approaches are available for chlorides that are broadly based on the error function solution to Fick's law [1]. Carbonation modelling based on a root time relationship is also available, but as the climate changes, this needs to consider the increase in carbon dioxide that is known to be occurring. Modelling deterioration due to aggressive ground is not commonly available, despite this being the main form of aggression that governs the concrete mix design for buried structures.

## 2. Modelling Chlorides

The long-term passage of chlorides into concrete is commonly assumed to be based on diffusion. There is an effect of surface absorption, but this is typically considered to only affect the outer surface. This process is typically modelled using the error function solution to Fick's second law, as shown below:

$$C_x = C_s \left( 1 - \mathrm{erf}\left( \frac{x}{2\sqrt{D_c \cdot t}} \right) \right) \tag{1}$$

where $C_x$ is the concentration at depth $x$, $C_s$ is the concentration at the surface, $x$ is the depth, $D_c$ is the diffusion coefficient, and $t$ is the time.

The surface concentration is normally extrapolated from a chloride profile rather than a value actually measured at the surface, and so it is often termed the effective surface concentration. The diffusion coefficient is generally considered to decrease over time, and so in turn, this is often termed the apparent diffusion coefficient. The equation can be used to determine a diffusion coefficient required from the concrete where the value of chloride at the depth of steel means that corrosion will not initiate for the given cover, as illustrated in Figure 1, based on a corrosion threshold of 0.4% by mass of cement, a surface concentration of 1%, and 100 years of exposure.

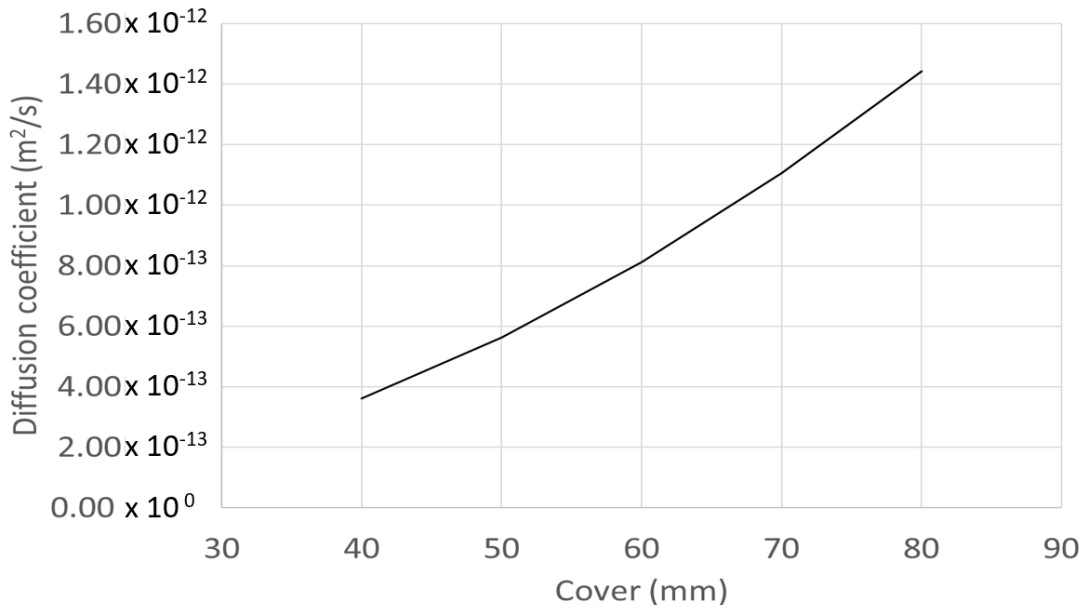

**Figure 1.** Example of the required diffusion coefficient for a given cover to prevent corrosion for 100 years, with a 0.4% corrosion threshold and 1% effective surface concentration.

In this example, the diffusion coefficient required at 40 mm cover is $3.6 \times 10^{-13}$ m$^2$/s. A cover of 70 mm requires a diffusion coefficient of $1.11 \times 10^{-12}$ m$^2$/s, approximately three times higher.

The problems arise when a more detailed review of the process is undertaken. Chatterji [2] stated that the assumption of constant diffusivity is seldom satisfied, with results suggesting that an increasing or decreasing depth of penetration could be observed depending on history. He adds that the use of a measured diffusion coefficient by any method to calculate the long-term chloride penetration depth is uncalled for and may be misleading.

Theoretically, Fick's second law cannot be used to analyse the chloride ion migration data through cementitious materials since no account is made for any chemical reactions that take place during the process. No account is taken of the Nernst relationship, which states that the diffusivity of an ion increases with increasing dilution, or of the formation of double layers on the cement hydration products. Chatterji [2] also stated that there is some evidence that in a steady state condition, the diffusion coefficient increases with a decreasing concentration of sodium chloride solution. Added to this, cement is a multi-component system containing many ions ($Ca^{2+}$, $Na^+$, $OH^-$, and $Cl^-$) and there is evidence that a single ion may move in two directions. Kondo et al. [3] reported that to maintain a charge balance, hydroxide ions diffuse out as chloride diffuses in. Saetta et al. [4] reported that temperature, relative humidity, and degree of hydration all have an effect on the effective diffusion coefficients. Dhir et al. [5] stated that from the properties of the initial mix, the diffusion coefficient at any time cannot be worked out and asks whether it is practical to quote the diffusion coefficient or time taken for a specific amount of chloride to reach the reinforcement for any practical situation since it varies with the environment. The authors further added that concrete specifications give no guidance as to the likely

durability of concrete since they specify by strength, cement content, or water/cement ratio, and these properties cannot ensure adequate durability.

Chatterji [2] concluded that since all results show very little evidence of a constant diffusivity, Fick's laws should not be used, even on an empirical basis. He further added that the use of an electrical field to accelerate the diffusion is unrealistic since cations and anions are forced to move in opposite directions, and this will interfere with the formation of Friedel's salt, and that work carried out in Norway supports these conclusions [6].

Two examples of readily available models in common use to relate the water/cement ratio to a diffusion coefficient are Life 365 from Ehlen et al. [7] and Bamforth [8]. Both have modifications for the use of cement replacements such as pulverised fuel ash (PFA) and ground granulated blast-furnace slag (GGBS), time dependence of the diffusion coefficient, and changes in surface concentration. There is a level of complexity in the maths that tends to make the user lose site of the natural variability that is found in chloride diffusion in concrete. The correlation between the water cement ratio and diffusion coefficient has been developed based on the data shown in Figure 2.

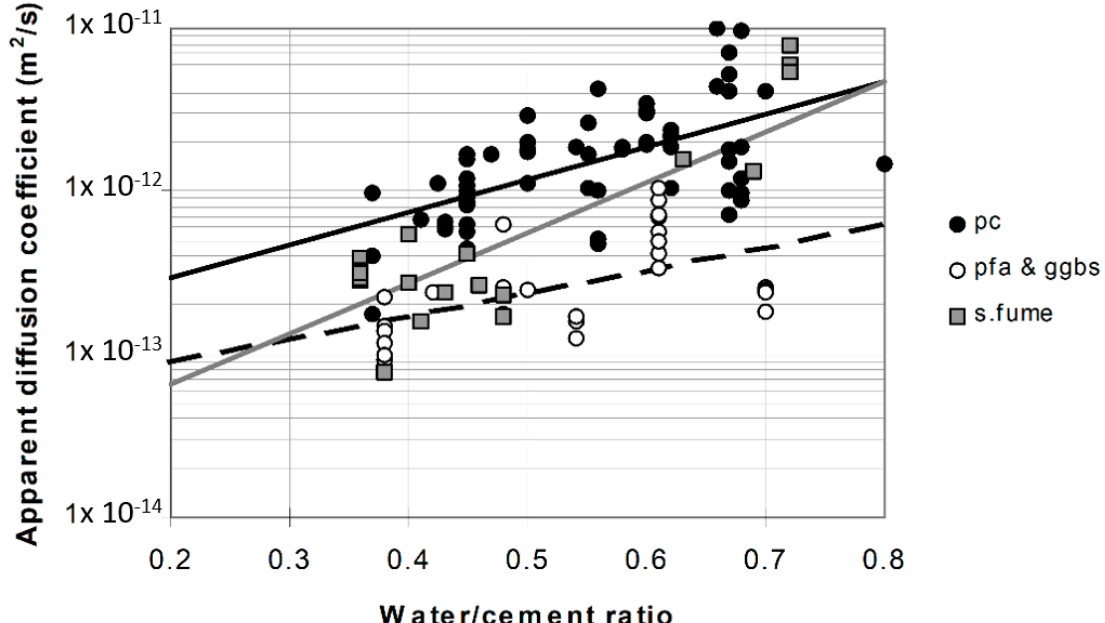

**Figure 2.** Diffusion coefficient vs. water/cement ratio, presented in Bamforth [8]. Adapted with permission from [8] Copyright 2004, The Concrete Society.

Bamforth stated that the variabilities in the calculated diffusion coefficients in laboratory samples are as high as 5 or 6 times, which needs to be considered when an increase by 2.5 times means that the cover in the above example goes from 40 mm to 70 mm, which is an excessive depth of cover that would not be acceptable in actual structures and would result in an increased risk of surface cracking. The spread of data for any given water/cement ratio is significant when compared with the comparatively narrow range used to define a required cover. We note that the cements used in developing the model are unlikely to include non-reactive fillers, such as calcium carbonate or limestone dust, that will affect the diffusion parameters and corrosion threshold. It is possible to develop a diffusion coefficient either through testing or from published correlations, but it is important to not lose sight of the variability associated with this figure.

In addition to the diffusion coefficient, it is necessary to include a figure for the surface concentration of chlorides.

The surface concentration used in the example is significantly lower than that found in corroding structures. Ehlen et al. [7] used a surface concentration of chlorides that increases with time to reach peak values. There are geographical variations due to the application of

de-icing salts, and there are also variations in the marine exposure conditions. The peak values for the marine conditions are presented in Table 1.

**Table 1.** Peak values for marine conditions (Data from Ehlen et al. [7]).

| Environment | Peak Surface Concentration (% by Weight of Concrete) |
| --- | --- |
| Marine splash zone | 0.8 |
| Marine spray zone | 1.0 |
| Within 800 m of the ocean | 0.6 |
| Within 1.5 km of the ocean | 0.6 |

Bamforth [8] suggested that different values depended on whether the concrete contains cement replacements. Table 2 reproduces the values presented by Bamforth [8].

**Table 2.** Peak values for effective surface concentration.

| Environment | Average Effective Surface Concentration (% by Weight of Concrete) | Characteristic Surface Concentration (% by Weight of Concrete) |
| --- | --- | --- |
| Portland cement-based concretes | 0.36 | 0.75 |
| Blended cement mixes | 0.51 | 0.9 |

It is worth noting that Bamforth [8] does not attempt to categorise the chloride-containing environment.

For a mix with 300 kg of cementitious material per cubic metre, the surface concentrations would be multiplied by eight to be comparable with the 0.4% corrosion threshold used above. Higher cement contents would reduce the multiplier. The peak surface concentration would therefore range from 2.4% by mass of cement outside of a marine splash or spray zone up to 8% by mass of cement, based on a 300 $kg/m^3$ cementitious material.

Repeating the calculation used in Figure 1, but with a surface concentration of 2.4% chloride by mass of cement, to achieve a 100 year life with 50 mm of cover, a diffusion coefficient of $2 \times 10^{-13}$ is required.

Reviewing the data from previous inspections undertaken by the authors for chloride surface concentrations extrapolated from chloride profiles, 2.4% is not especially high, but it should be noted that most inspections are undertaken on deteriorating structures where corrosion has already started.

The only environment where one would expect the surface concentration to be easily defined is in submerged structures but comparing the concentration of solutions in mg/L with the required percentage of chloride by mass is not straightforward. At first glance, the chloride concentration in the water outside the concrete would be expected to be identical to the chloride concentration inside the concrete, at equilibrium. For example, for 20 g of chloride per litre of water outside the concrete, once equilibrium is reached, there should be 20 g of chloride per litre of concrete inside. Assuming a density of concrete of 2400 $kg/m^3$, this should equate to 0.8% by mass of concrete, but this is not the case. A proportion of the aggregate may or may not be porous, and so it will not be part of the volume that can reach equilibrium. This should result in the concentration inside the concrete being lower than that outside. Cementitious materials, including cement replacements such as GGBS or PFA, will bind an amount of the diffused chlorides, which would lead to the equilibrium level potentially having higher chlorides inside the concrete than outside. It should also be noted that activity is the actual driving force behind most chemical processes considered to be dominated by concentration. Activity is a dimensionless ratio that is a function of the concentration multiplied by an activity coefficient. For diluted solutions, the activity coefficient is typically assumed to be unity. Concrete pore solutions are not diluted. Published works have suggested that the activity coefficients in cement paste are of the order of 0.46 [9]. This could mean that the equilibrium level could be up to 2.2 times higher than the level in the external solution, but we note that the activity coefficient of the

external solution is unlikely to be unity. The data in Table 3 is extracted from Potter [10], where 20 g/L Cl of a solution based on sodium chloride would be expected to have an activity coefficient of approximately 0.66, and so would behave as if it were a 13.2 g/L solution. This could equilibrate with the internal pore solution if the activity internally was equivalent to 13.2 g/L of concrete. Employing an activity coefficient of 0.46, this would be at a concentration of 29 g/L of concrete or 1.2% by mass of concrete. It should be noted that the data in Table 3 are the mean ionic activity coefficients (not specific coefficients for an ion). It is possible to theoretically calculate and experimentally determine the activity coefficients (or the activity) of the chloride ion in synthetic or real cement paste pore solutions [11,12], but it is not commonly undertaken outside of a laboratory.

**Table 3.** Activity coefficients for various chemical species.

| Concentration (M) | HCl | $H_2SO_4$ | NaCl | NaOH |
|---|---|---|---|---|
| 0.1 | 0.796 | 0.265 | 0.778 | 0.766 |
| 0.5 | 0.757 | 0.154 | 0.681 | 0.690 |
| 1 | 0.809 | 0.130 | 0.657 | 0.678 |
| 2 | 1.009 | 0.124 | 0.668 | 0.709 |

The work by Lindvall [13] showed that the effective surface concentration in samples exposed to 20 g/L was approximately 3.5% by mass of cement. The samples were made using 450 kg/m$^3$, which equates to approximately 0.63% by mass of sample and is significantly lower than the 1.2% by mass of sample calculated above. The work also exposed samples to 5 g/L, which produced an effective surface concentration of approximately 2% by mass of cement or 0.38% by mass of sample. A sodium chloride solution of 5 g/L would be expected to have an activity coefficient of 0.76; based on the differences in activity coefficient, this would be expected to equilibrate at 0.34% by mass of sample, which is similar to that found by Lindvall. It is possible that the activity coefficient of the 20 g/L external solution was lower than that estimated as the counter diffusion of ions from the concrete into the solution would have an effect that may be more pronounced in the more concentrated solution. The concrete samples were the same types, and so it is reasonable to assume that both sets had similar proportions of porous material.

It should be noted that if structures are hollow and have a dry interior, evaporation of ingressing moisture on the internal face will lead to a significant localised increase in surface concentration that will become difficult to manage within a model. As the evaporation progresses, the chloride concentration would tend towards a saturated solution. In practice, effective surface concentrations over 8% by mass of cement are rarely encountered, based on the author's experience, and so it may be appropriate to use the peak figures found in Life 365 as a natural upper limit. In these cases, it may be better to include for the inspection and maintenance of isolated leaks than to design all the concrete to be able to withstand the extremely high levels of chloride that can be found.

All of the above ignores the fact that the amount of chloride required to initiate the corrosion of steel in concrete is not precisely defined, even though the topic was raised as early as 1967 by Hausmann [14]. The value most easily measured is the total chloride content of a sample, which can then be converted into a mass of chloride by mass of cement, if required. This conversion often assumes the cement content of the mix. It is possible to extract a dust sample for a measurement of cement content, but the tests in common usage measure the total calcium or silica content and are complicated by the presence of aggregate that contains calcium or silica. In addition, the sample sizes often collected are too small to accurately represent concrete. Round robin testing undertaken in the UK reported that 1 in 20 reproducibility values from tests undertaken to BS 1881 Part 124 [15] are of the order of +/− 160 kg/m$^3$ [16]. There are alternatives [17], but these methods are yet to be covered under an international standard and so are not in common usage. The 1 in 20 reproducibility figures reported for chloride analysis were up to 0.097% by mass of sample.

The total chloride content will contain chlorides free to contribute to the corrosion processes, loosely bound chlorides that may be liberated as internal equilibrium change, and chemically bound chlorides that are typically considered to not contribute to the corrosion process [18].

Life 365 [7] uses a single value of 0.05% by mass of sample and Bamforth [8] uses 0.4% by mass of cement, with adjustments for cement replacements and temperature, as a basis for design erring on the side of caution. Both documents highlight that there is a range of values that could be used, and it would be better to use a risk-based approach. Alonso et al. [19] reported a threshold from 0.097 to 3.04% by weight of binder. Oh et al. [20] reported a total chloride threshold of between 1.24 and 3.08% by mass of binder. Stratfull et al. [21] reported a range of between 0.68 and 0.97% by mass of binder. Troconis de Rincon et al. [22] concluded that elevated temperatures reduced the corrosion threshold, with exposure at a marine tropical location having a chloride threshold of approximately 0.42% by mass of cement and a non-tropical site having a threshold of approximately 0.89% by mass of cement. The reported chloride thresholds by mass of concrete range from 0.03 [23,24] to 0.2 [25].

Other available models such as the Duracrete model [26] use probabilistic distributions, with marine submerged zones using a mean of 2% by mass of binder and a standard deviation of 0.15 and atmospheric zones using a mean of 0.6% by mass of binder and a similar standard deviation.

To summarise the above, it is possible to develop a mathematical approach and develop a diffusion coefficient and surface concentration to allow for modelling chloride penetration into concrete, but the variability associated with both these figures and the amount of chloride required to initiate corrosion, and hence the end result, need to be fundamentally understood for this to be effective.

### 3. Modelling Carbonation

Carbonation is generally considered to follow a root–time relationship. Typically, it is assumed that the relationship does not change over the period being modelled, but there is data available that suggests this is not the case [27]. Unlike chloride penetration, carbonation is likely to be significantly affected by climate change as the levels of atmospheric carbon dioxide increase significantly. Bamforth [8] included a carbonation model that includes an option for increasing carbon dioxide concentrations. The model is called Carbuff and is based on the ability of concrete to react with carbon dioxide in a process known as buffering. It provides a calculated typical life, or design life, based on carbonation depth reaching the steel, and a time-to-cracking based on reinforcement corrosion. Essentially, it assumes that the depth of carbonation is primarily a function of the amount of Portland cement contained within a mix, and, more specifically, is a function of the tricalcium aluminate content of that cement.

The main problem with this approach is that the use of cement replacements reduces the amount of cement in a given mix, and in Carbuff, this significantly reduces the performance of the concrete. The model does not consider the reduced diffusion coefficient associated with the use of cement replacements, the enhanced resistance commonly assumed to be achieved by an increase in strength or reduction in water/cement ratio, or the reduction in corrosion rates associated with the increased electrical resistance found when using cement replacements.

If a comparison is made between the model and the current European codes, such as EN 206 [28], mixes that achieve a 50 year life with Portland cement will only achieve a 21 year life if cement replacements are used, and experience shows that this is not the case. For a 100 year life, the cover and cement combinations with Portland cement only achieve 75 years, and, again, cement replacements reduce this further.

Bamforth [8] acknowledged the model is conservative and stated:

"the predicted rates of carbonation are, on average, about 40% higher than measured values, as shown in Figure 10.17, with almost all predicted values exceeding reported carbonation rates."

Figure 10.17 is reproduced here as Figure 3.

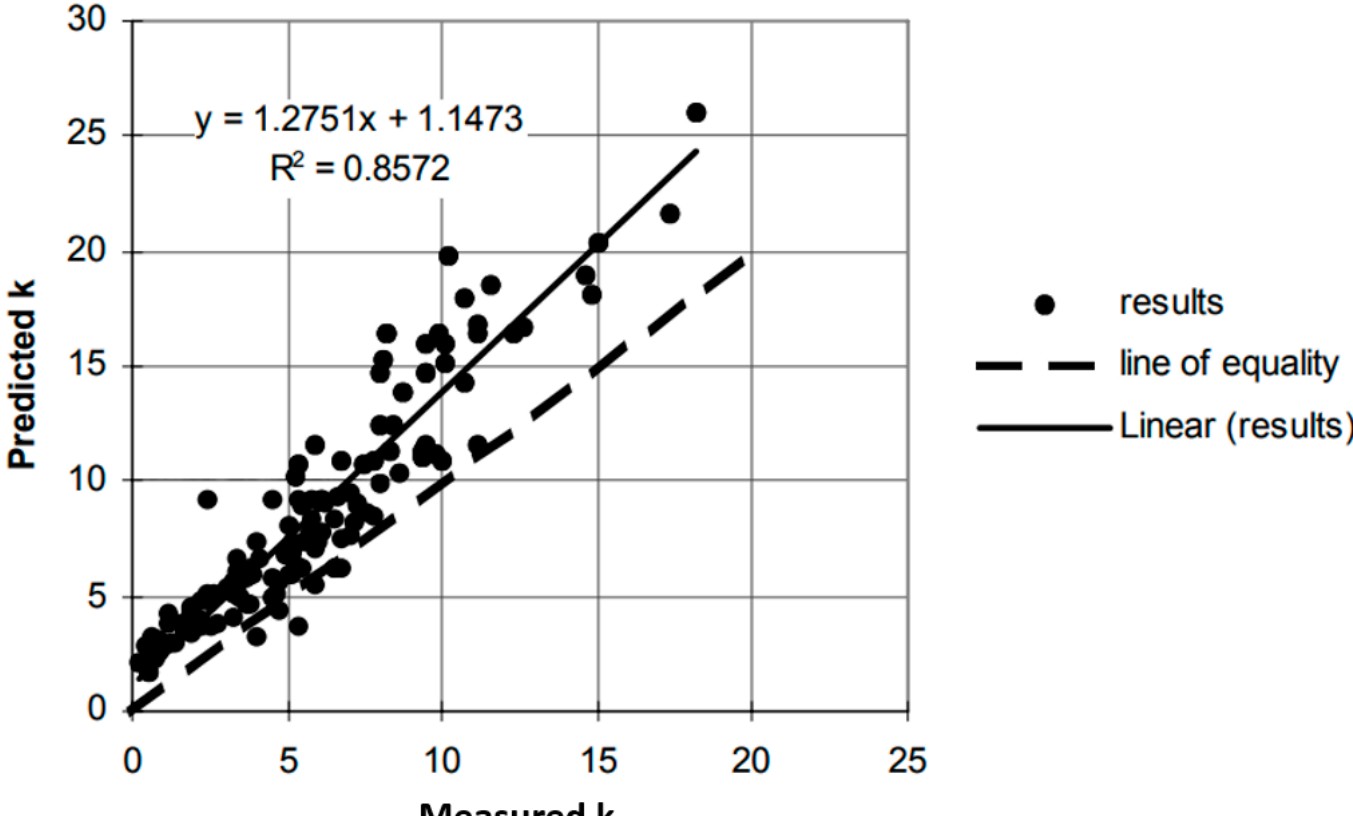

**Figure 3.** Predicted k vs. measured k, from Bamforth [8]. Adapted with permission from [8] Copyright 2004, The Concrete Society.

The 40% figure quoted relates to a coefficient that is multiplied by the square root of time to calculate a carbonation depth. We note that the data used came from a number of studies that were published prior to 1990. Data are also available from a road tunnel study that was subsequently published [29], showing the effects of an average 600 ppm environment and 1200 ppm environment. The 600 ppm environment accelerated carbonation by 20%, and the 1200 ppm environment accelerated carbonation by 70%. By comparison, Carbuff increases the rate by 40% and 100%, respectively. This is not the fault of the model since the data from the road tunnels was published in 2019. It is not suggested there is a problem with using old data, but cement production, mix designs, placement, and curing change over time. In addition, carbon dioxide levels are changing. The Fifth Assessment Report of the Intergovernmental Panel on Climate Change (IPCC) [30] provided Figure 12.36, shown below as Figure 4.

If a structure is being built today and has a 100 year life, it will be exposed to a significantly higher $CO_2$ level. The predicted carbon dioxide levels depend on how much we manage to reduce emissions.

Carbuff allows the modeller to increase the carbon dioxide levels in exposure, and this produces a corresponding increase in the rate of carbonation. It is a logical approach, but data are available from structures operating in an increased level of carbon dioxide. From Carbuff, picking a typical mix and 350 ppm of $CO_2$ in the atmosphere, 40 mm of cover will typically corrode in 50 years. At 700 ppm, the same 40 mm of cover will corrode in only 25 years, and at 900 ppm, it will corrode in 20 years. The cover needed to provide the

same 50 year life at 700 and 900 ppm of $CO_2$ is 55 and 65 mm, respectively. If you consider 700 ppm $CO_2$ levels at 75% relative humidity, the modelled cover required for a 450 kg/m$^3$ cement content mix with a required design life of 100 years before corrosion starts is as shown in Table 4.

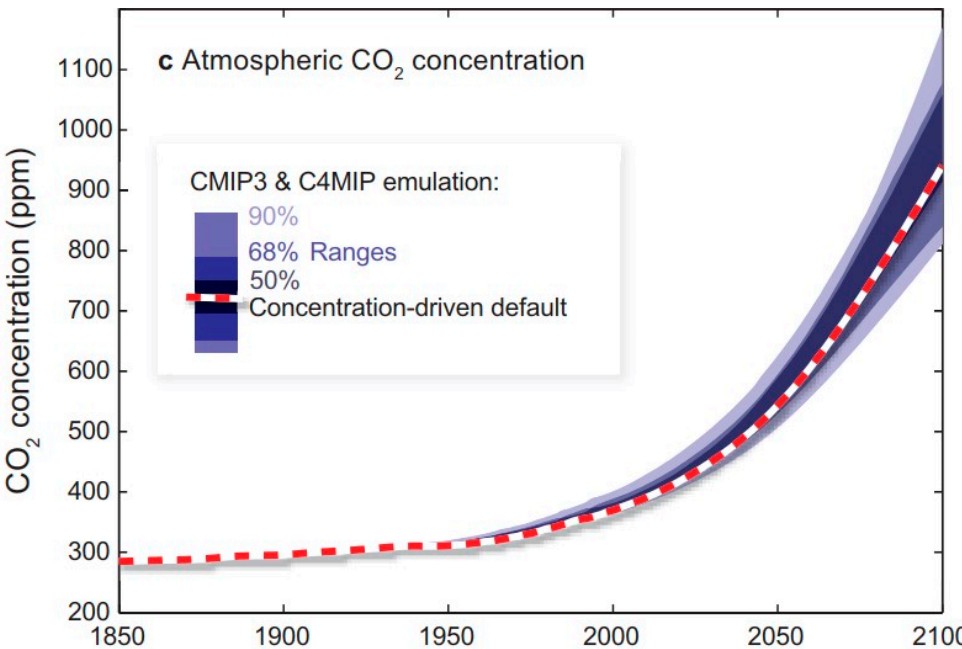

**Figure 4.** IPCC-predicted atmospheric carbon dioxide levels from IPCC 2013, Figure 12.36, Panel (c) [13]. Adapted with permission from Ref [13] Copyright 2013, The Intergovernmental Panel on Climate Change.

**Table 4.** Cover required for 100 year life with 450 kg/m$^3$ cementitious material at 75% RH (700 ppm $CO_2$).

| Concrete Mix. | Carbuff Design Cover | Carbuff Typical Cover Required for Carbonation | Carbuff Cover Reduced K to Reflect Figure 3 |
|---|---|---|---|
| Portland cement-based concretes | 37 | 45 | 27 |
| 30% PFA | 62 | 75 | 40 |
| 70% GGBS | 57 | 67 | 36 |

If following the designed cover from the unmodified model, the conservatism means that an additional 20–25 mm of cover is required if cement replacements are used.

A further point to consider is that the pH reduction associated with carbonation is almost certainly going to reduce the corrosion threshold. Confirming that a mix is acceptable for carbonation and chloride ingress as two separate exercises may not produce a reliable answer.

## 4. Sustainability

The general consequences of attempting to produce durable concrete are to drive down water cement ratios and increase cover. At a high level, this will increase durability. The lower water cement ratios will also produce a need for higher cement contents to maintain workability. The increased cover required will necessitate an increase in the mass of concrete used. This will have a significant effect on the embodied carbon of a mix.

Figure 5 shows the embodied carbon equivalent for the cement component of a 250 thick 1 m × 1 m panel at a range of cementitious material contents, including data for the replacement of PC with 70% ground granulated blast furnace slag (GGBS) or

30% pulverised fuel ash (PFA). We note that the embodied carbon figures for the materials will vary depending on the source and on the carbon used in producing the power required to manufacture the product. The embodied carbon associated with the reinforcement steel is not included. The cement replacements illustrated are common cement replacements, but supply is becoming limited. Typically, the products are either by-products of coal-fired power stations or of the blast furnace slag production of steel. Coal-fired power stations need to be reduced to manage climate change. Blast furnaces are an energy-intensive method of producing steel and are becoming less common. Cement replacements affect the workability of the mix, and so either the water/binder ratio is increased or the binder content is increased, which offsets the sustainability benefits if there ends up being a need for increasing the cement contents. Higher cement contents also increase the heat generated, which in turn increases the demand for more steel in the concrete to control thermal cracking. This is not considered in the figure.

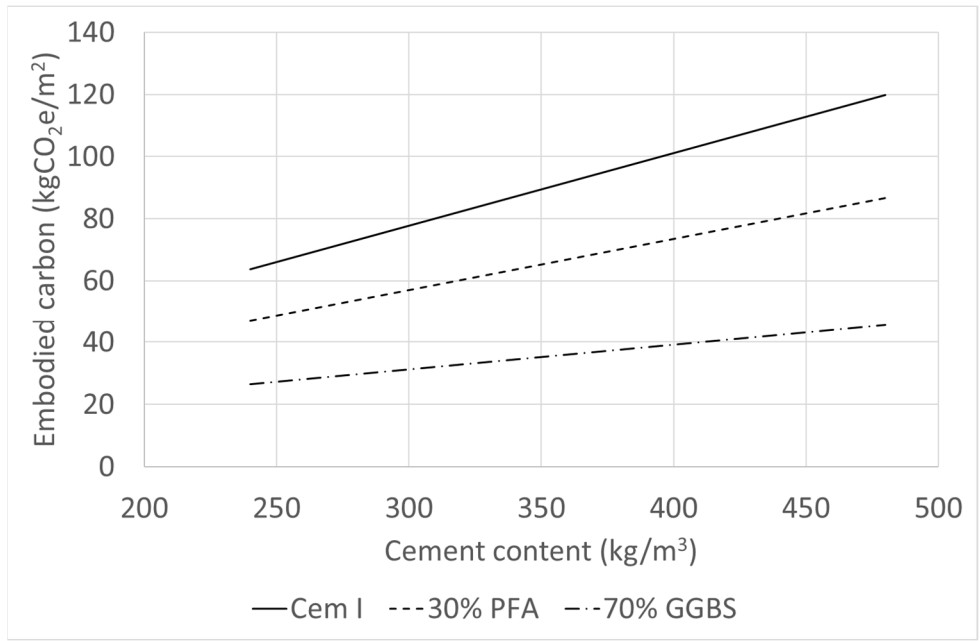

**Figure 5.** Embodied carbon in a 250 mm thick 1 m × 1 m panel for different cement/binder combinations.

Slag produces the lowest $CO_2$ emissions option because it replaces more of the cement; however, to achieve the required durability due to carbonation based on Carbuff, more cover is required.

Based on Table 4, PFA requires 13–30 mm more cover than PC to resist carbonation and slag cement needs 9–22 mm more cover, which would be needed on both sides of the element. If this equates to a direct increase in element thickness over the 250 mm thickness, another 10 to 20% more concrete will be needed for pfa and 7 to 17% for GGBS, which means the benefits of using cement replacements are not straightforward. GGBS is still likely to produce the lowest embodied $CO_2$ option, but PFA tends towards Cem I values in direct benefits, noting that the heat of hydration would be lower so that the thermal steel requirements would still save embodied carbon.

The embodied carbon for repairs has been reviewed by Atkins and Lambert [31]. This work concluded that the most carbon-efficient method of addressing durability is to prevent the need for repair, either by the inclusion of coatings or cathodic protection. Installing cathodic protection at the time of construction would add approximately 4 kg $CO_2$ e/m$^2$ of concrete. This can be compared with the carbon equivalent of a change in the mix design or cover, as shown in Figure 6.

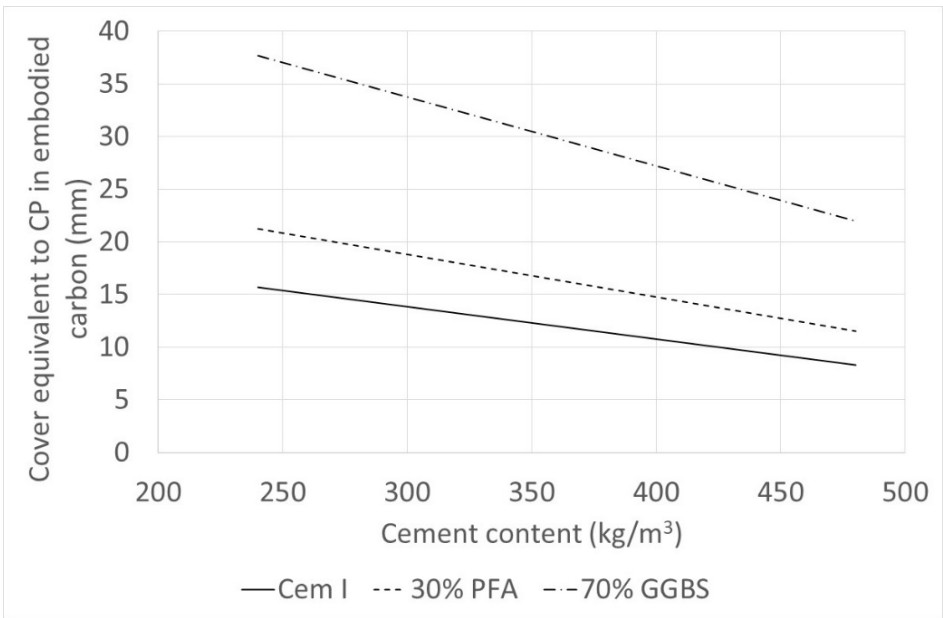

**Figure 6.** Cover equivalent (in mm) to the embodied carbon of a cathodic protection system.

The embodied carbon in coating concrete has been calculated as 2.7 kg $CO_2$ e/m². The coating option would need to be applied multiple times throughout the life of the structure, but it could only be applied once to reduce the overall duration of atmospheric exposure. This would only be possible in accessible areas, but the recoating could be targeted in areas that are subject to degradation. Figure 7 presents the cover required to achieve a given life based on a root–time relationship.

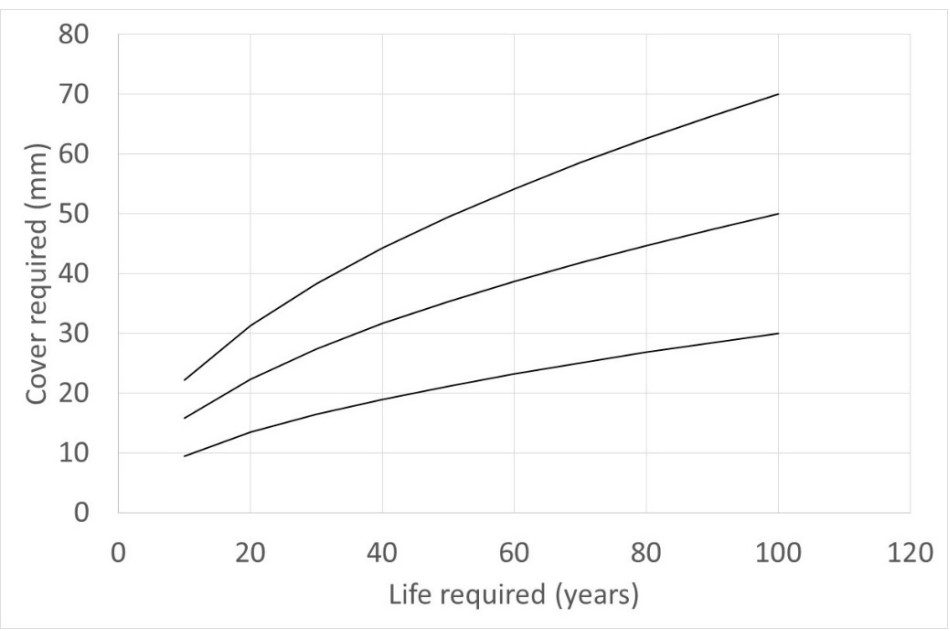

**Figure 7.** Cover (in mm) to achieve a given life.

If, for example, 70 mm of cover would be required to achieve a 100 year life and a coating is employed with a life of 20 years, the required cover would be that needed for an 80 year life, which is 63 mm. The embodied carbon of the extra 7 mm of concrete can then be compared with the embodied carbon in a coating. This is shown in Figure 8.

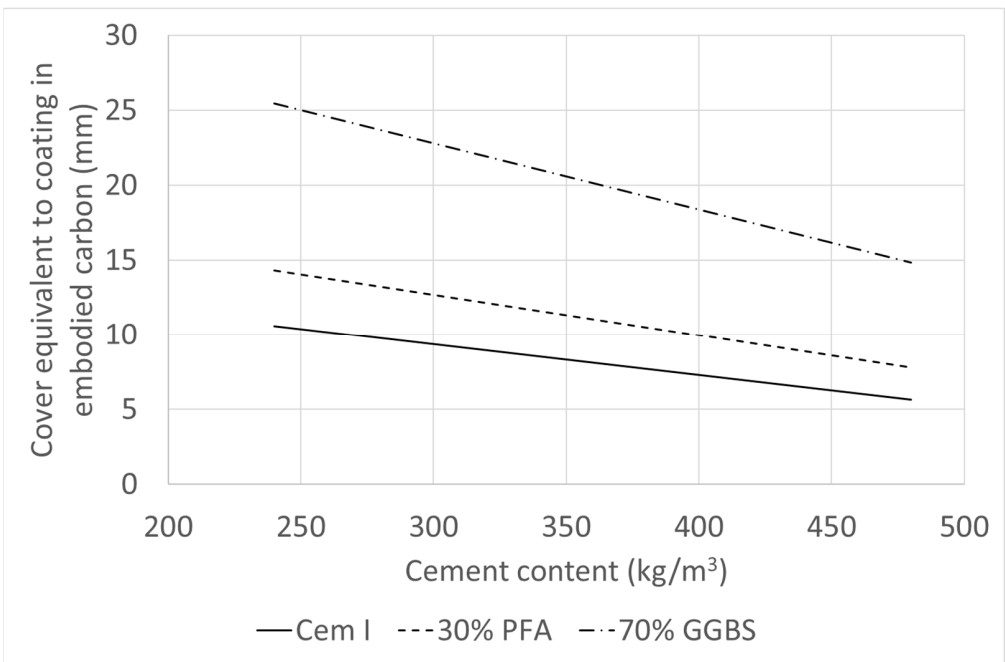

**Figure 8.** Cover equivalent (mm) to the embodied carbon in a coating.

Taking the 7 mm of cover that could be saved with a coating, this would not result in a saving of embodied carbon for a 70% GGBS mix, but it would result in a carbon saving for Cem I with greater than approximately 350 kg/m³ of binder or a 30% PFA mix with greater than 450 kg/m³.

The decision to recoat at periodic intervals would require planned maintenance to be included at the design stage. This targeted approach of actively planning maintenance is the approach taken for steel structures but is generally avoided for concrete. If maintainability were planned, the natural variability in future predictions, for example, on climate change, or time-dependent properties of concrete could be lessened. Significant sustainability benefits could be achieved by including the ability to maintain concrete structures in the design, rather than trying to ensure the concrete mix compensates for variable future scenarios.

## 5. Other Deterioration Mechanisms

While steel corrosion in concrete is considered the most common form of degradation of reinforced concrete, the concrete matrix itself can degrade, and there are few models available to address this. Acidic soils will attack cement. In addition, carbon dioxide dissolved in water will attack the cement itself, and as the atmospheric levels increase, this is likely to become more of an issue.

Sulfate attack is a key parameter in mix design and there are limited approaches to modelling this. The *BRE Special Digest 1* [32] provides guidance on increasing concrete cover to allow for sulfate attack, but as the publication explains, it is based on a limited investigation on thaumasite-attacked structures, where the attack depth was up to 50 mm in 30 years, with a concrete mix design that would not be expected to be durable in the exposure conditions found. It recommends that an additional thickness of 50 mm would be adequate for concrete with an intended working life of 50 years, but this should not be used where a sulfate attack could affect the structural integrity, for example, by causing expansion forces or reducing friction.

## 6. Conclusions

There are some common degradation mechanisms that affect reinforced concrete, and a number of models have been produced that attempt to capture these. The detail in the equations suggests a precision in the answers that may not adequately reflect the variability

found in key parameters. In addition, there are other degradation mechanisms that occur that are not typically modelled. When producing a durability-based design, it is important to not lose sight of the data that are used behind the models.

Durability requirements need to be balanced alongside sustainability requirements if construction is to contribute to reducing carbon dioxide emissions. Planned routine maintenance for concrete should provide significant sustainability benefits.

**Author Contributions:** All authors have read and agreed to the published version of the manuscript.

**Funding:** This research received no external funding.

**Institutional Review Board Statement:** Not Applicable.

**Informed Consent Statement:** Not applicable.

**Data Availability Statement:** Not applicable.

**Conflicts of Interest:** The authors declare no conflict of interest.

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
