# Peer review of "Limitations in Modelling Reinforced Concrete Durability"

_cmd, doi:10.3390/cmd3030019_

Round 1
Reviewer 1 Report
The conrete cover cannot be increased arbitrarily. I would not see any promblems in the concrete compression zone. In the concrete tension zone, the width will inevitably increase, since there is a proportional increase from the elongation of the steel. Concrete coverings higher than 5 cm require additional reinforcement. This should be explained by the authors, when they describe the influence of the concrete cover on the service life.
Author Response
Dear Reviewer,
We thank you for your helpful observations and have amended the manuscript to better explain the role and limitations of changes in cover depth. This amendment has been highlighted in the text.
Regards,
Chris Atkins & Paul Lambert
Reviewer 2 Report
-The manuscript cannot be accepted in the "corrosion and materials degradation" There are many mistakes, insufficient explanations and vague sentences. Thus, I must reject it.
Author Response
Dear Reviewer,
We acknowledge your comments but have updated our manuscript based on the comments received from the Peer Reviewers.
We trust that in its updated form it will be considered suitable for publication.
Regards,
Chris Atkins & Paul Lambert
Reviewer 3 Report
Some specific comments should be considered for improving the quality of manuscript.
- Abstract need to well organized, which should address critical review conducted on the methods, major results, propose methods that limit the current methods in addressing the limitations of Modelling Reinforced Concrete Durability. Demonstrate in the abstract novelty and practical significance for the proposed work.
- The introduction session needs to be well reorganized. In the introduction section, the authors mentioned the existence of previous studies. However, critical review of methods are completely missing. The parameters that critically influence the modelling need to be well addressed. Critical review of individual parameters needs to be highlightened.
- Authors need to provide the flowchart of different modelling approaches and highlight their advantages in the table. If the Figure 1 has been retrieved from any references kindly provide.
- Highlight the cost-effective experimental methods for developing models, empirical relationships that could provide sufficient input-output data for analysis.
- It would be better to highlight the details of artificial intelligence modelling tools that could provide extended benefits.
- Authors need to highlight the reverse modelling using ANNs and their significance in practical cases.
- Critical review on past studies of ANNs and artificial intelligence in optimizing the concrete durability with process parameters need to be highlightened.
- The range for Carbuff Design cover, Carbuff Typical cover required for carbonation, and Carbuff Cover Reduced K to reflect figure 10.17. need to be highlightened. Similarly, for other table parameters.
- Conclusion should be highlightened with a focus on critical review conducted and objective conclusions drawn in bulletin points.
- Furthermore, scope for future work need to be highlightened.
- Authors need to revise manuscript with recent literatures. There are very limited references are used, which clearly dictates a brief review rather than critical review.
Author Response
Dear Reviewer,
We are grateful to receive you detailed comments and recommendations for improvement of our manuscript.
These have been accommodated as far a possible in our updated version which we hope will be considered worthy of publication.
With respect to the use artificial intelligence and reverse modelling with artificial neural networks, this is outside our area of knowledge and experience and we do not feel able to incorporate comment about their possible role in a sufficiently informed manner.
We trust you will understand.
Regards,
Chris Atkins & Paul Lambert
Reviewer 4 Report
Line 20: You use the term "embodied carbon" through all the text. I guess that you mean "embodied carbon to environment". I think clarity will be enhanced adding "to environment" at least in this first appearance of the term.
Lines 44 and 46: X and T should be lowercase (x and t) according to equation in line 40.
Figure 2: Symbol for pfa & ggbs is missing in the legend.
Lines 110 and 126: Number 3 in the units should be a superscript.
Table 4: Header in last column refers to figure 10.17. I think it should refer to figure 3 instead, because the figure is labeled in this way in this paper.
Author Response
Dear Reviewer,
Thank you for you valuable feedback.
We have responded to your comments and amended the manuscript accordingly.
Regards,
Chris Atkins & Paul Lambert
Reviewer 5 Report
The manuscript presented gives a discussion of some limitations authors have observed from several durability modelling of concrete structures. The following are some concerns that the authors should take into consideration to improve the manuscript:
- Figure 2 shows some missing data points in, for example, w/c=0.6 and apparent diffusion coefficient (Dap) between 10-13 and 10-12. Please, make the corrections so all data points appear in the graph. It happens also with the missing symbol of pfs & ggbs, at the right of the graph.
- Interesting discussion about chloride surface concentration (Cs) and threshold (Ct). There is no comment regarding the concrete´s porosity effect on both values Cs and Ct.
- No calculations mentioned included possible chemical absorption of chlorides, i.e. ggbs chemically bounds more chlorides than any other cementitious compound in concrete.
- What about exposure temperature on the diffusion of chlorides in concrete? The references used include natural exposure in temperate climate, what about temperatures at tropical marine environment? Values discussed should be reconsidered or at least put some warnings that were obtained in climates with temperate environments. In reference [1] are some discussions regarding the effect of temperature on Ct.
- Another topic not addressed in this manuscript is the fact that actual cement formulations include not only puzzolans, but also non-reactive fillers, i.e., calcium carbonate or limestone. This filler additions has being proved to decrease Ct and Dap, thus all the modeling defined at this moment with regular Portland cement (>90% clinker content) will work for structures constructed before year 2000 (when filler started to being used). Authors need to comment on this as other limitations on actual durability models.
- Now regarding carbonation, data presented in figure 3 shows similar performance in other publications with natural exposure results after several years of exposure: predicted K values are overestimated, compared to measured K values [2].
- Another limitation on carbonation estimates using the square root of time (SRT) model is the effect on urban and marine environments. Chlorides are hygroscopic and help to maintain humid concrete porosity. This also reduces the carbonation chemical process and/or transport of CO2 gas in concrete.
- In reference [2] there is a discussion on the carbonation depth of concrete exposed to natural environment, instead of accelerated methods: there is a steady increase over two years followed by a gradual leveling out after 3-5 years exposure. This was observed after 2-4 years of natural exposure; thus, carbonation did not follow the SRT approach. Are the authors familiar with this phenomenon?
- As stated in (5), the commercialization of supplementary cementing materials (SCM) or limestone fillers, to decrease CO2 emissions due to clinker production, decreases the alkalinity of concrete and increases the carbonation of same concrete. This of course, decreases durability of concrete, either on chloride and CO2 laden environments, thus sustainability of concrete structures due to earlier repairs or substitutions disappear need to use more clinker for repair materials in the failed structures, thus saving clinker at the beginning is not entirely correct for reducing carbon dioxide emissions considering the entire life cycle of the concrete structure. Do authors have any comments regarding the sustainability of concrete structures when such clinker decrease is affecting the durability of the structures and thus its sustainability?
References
[1] Troconis de Rincón, O., et al. (2016). “Reinforced Concrete Durability in Marine Environments DURACON Project: Long-Term Exposure,” Corrosion, V.72, No.6, pp. 824-833, ISSN: 0010-9312.
[2] Troconis de Rincón, O., et al. (2015). “Concrete Carbonation in Ibero-American Countries DURACON Project: Six-year Evaluation,” Corrosion, V.71, No.4, pp. 546-555, ISSN: 0010-9312.
Author Response
Dear Reviewer,
Thank you for taking the time to provide such detailed comment, which we are grateful to receive.
We have updated our manuscript accordingly.
Two exceptions are point 2 which we believe is addressed in lines 134 - 135, and point 9 which is covered in lines 309 to 311.
We trust you will find our responses sufficient and the manuscript now worthy of publication.
Regards,
Chris Atkins & Paul Lambert
Round 2
Reviewer 2 Report
-The manuscript is still containing many mistakes, insufficient explanations and vague sentences. Thus, I must reject it.
Author Response
Dear Reviewer,
We acknowledge your comments and have further responded to the recommendations of the reviewers and editors and now hope that the manuscript is sufficiently improved to warrant publication.
Best Regards,
Chris Atkins and Paul Lambert.
Reviewer 3 Report
Authors attempted to answer all quieries raised in the first review process. Therefore, possible acceptance can be ensured
Author Response
Dear Reviewer,
We are grateful for you helpful comments and suggestions and appreciate your time in checking they have been accommodated adequately within the updated manuscript.
Best Regards,
Chris Atkins and Paul Lambert
Reviewer 5 Report
Recommendations were included in the 2nd version. Good luck with the publication and future citations.
Author Response
Dear Reviewer,
We are very grateful for you helpful comments and suggestions and appreciate your time in checking they have been accommodated adequately within the updated manuscript and helping us bring it up to the required standard.
Best Regards,
Chris Atkins and Paul Lambert